# Parasitic insect-derived miRNAs modulate host development

Zhi-zhi Wang[1,2], Xi-qian Ye[1,3], Min Shi[1,2], Fei Li[1,2], Ze-hua Wang[1,2], Yue-nan Zhou[1,2], Qi-juan Gu[1,2], Xiao-tong Wu[1,2], Chuan-lin Yin[1,2], Dian-hao Guo[1,2], Rong-min Hu[1,2], Na-na Hu[1,2], Ting Chen[1,2], Bo-ying Zheng[1,2], Jia-ni Zou[1,2], Le-qing Zhan[1,2], Shu-jun Wei[4], Yan-ping Wang[1,3], Jian-hua Huang[1,2], Xiao-dong Fang[5], Michael R. Strand[6] & Xue-xin Chen[1,2,3]

Parasitic wasps produce several factors including venom, polydnaviruses (PDVs) and specialized wasp cells named teratocytes that benefit the survival of offspring by altering the physiology of hosts. However, the underlying molecular mechanisms for the alterations remain unclear. Here we find that the teratocytes of *Cotesia vestalis*, an endoparasitoid of the diamondback moth *Plutella xylostella*, and its associated bracovirus (CvBV) can produce miRNAs and deliver the products into the host via different ways. Certain miRNAs in the parasitized host are mainly produced by teratocytes, while the expression level of miRNAs encoded by CvBV can be 100-fold greater in parasitized hosts than non-parasitized ones. We further show that one teratocyte-produced miRNA (Cve-miR-281-3p) and one CvBV-produced miRNA (Cve-miR-novel22-5p-1) arrest host growth by modulating expression of the host ecdysone receptor (*EcR*). Altogether, our results show the first evidence of cross-species regulation by miRNAs in animal parasitism and their possible function in the alteration of host physiology during parasitism.

[1] Institute of Insect Science, College of Agriculture and Biotechnology, Zhejiang University, 310058 Hangzhou, China. [2] Ministry of Agriculture Key Lab of Molecular Biology of Crop Pathogens and Insect Pests, Zhejiang University, 310058 Hangzhou, China. [3] State Key Lab of Rice Biology, Zhejiang University, 310058 Hangzhou, China. [4] Institute of Plant and Environmental Protection, Beijing Academy of Agriculture and Forestry Sciences, 100097 Beijing, China. [5] BGI-Tech, BGI-Shenzhen, 518083 Shenzhen, China. [6] Department of Entomology, University of Georgia, Athens, GA 30602, USA. These authors contributed equally: Zhi-zhi Wang, Xi-qian Ye, Min Shi, Fei Li, Ze-hua Wang. Correspondence and requests for materials should be addressed to X.-x.C. (email: xxchen@zju.edu.cn)

Parasitic insects, particularly parasitic wasps, are a large group of animals that lay eggs in or on the bodies of other arthropods that serve as hosts for offspring development. More than 20% of all insect species are parasitic wasps, making them among the most species-rich animal groups on Earth[1]. Successful development of wasp offspring usually results in death of the host. As such, many parasitic species are used as biological control agents for management of pest species. The moth *Plutella xylostella* (order Lepidoptera: family Plutellidae) is a worldwide pest of cruciferous plants[2] and its larval stage is the natural host of the endoparasitic wasp *Cotesia vestalis* (family Braconidae). Many parasitic wasps kill their hosts by producing effector molecules that disable growth and immune defences. When laying eggs, *C. vestalis* introduces into *P. xylostella* effectors that derive from three sources: (1) venom produced in a venom gland, (2) a symbiotic polydnavirus named *C. vestalis* bracovirus (CvBV), which expresses virulence genes after infecting host cells, and (3) cells called teratocytes that are produced during embryogenesis and released into the body of the host when wasp eggs hatch[3–6].

Most known effector molecules produced by *C. vestalis* and other parasitic wasps are proteins. However, eukaryotes including insects and some viruses produce microRNAs (miRNAs) that are small (20–22 nts) non-coding RNA (snRNA)[7]. miRNAs regulate a range of cellular processes by repressing translation or promoting the decay of target transcripts through imperfect pairing with complementary sites in 3′-untranslated regions (UTRs)[7]. Several intracellular pathogens promote their own survival by modulating the miRNAs produced by hosts[8]. Some non-viral microbial pathogens also produce snRNAs that target host genes, while some hosts produce miRNAs that affect pathogen gene function[9–11]. However, it is unknown whether parasitic animals such as insects produce miRNAs that affect hosts.

Recent sequencing of the *C. vestalis* and *P. xylostella* genomes provide the references needed for identifying *C. vestalis*-produced miRNAs and assessing their roles in parasitism of *P. xylostella*. Here, we report that *C. vestalis* teratocytes express several miRNAs and CvBV encodes several miRNAs that are expressed in infected host. We further show that one teratocyte-produced miRNA and one CvBV-produced miRNA arrest host growth by modulating expression of the host ecdysone receptor (*EcR*).

## Results

***Cotesia vestalis* encodes multiple predicted miRNAs.** snRNA libraries were constructed from *C. vestalis* larvae and teratocytes. Sequencing yielded 12 (larval) and 15 million (teratocyte) raw reads, of which 74.6% mapped to the *C. vestalis* genome after quality filtering. In both libraries, read distributions showed two peaks at 22–23 nt and 28–29 nt, a typical pattern in most insect small RNA libraries (Supplementary Fig. 1)[12]. In total, 38% of the mappable reads in the *C. vestalis* larval libraries and 10.2% of the mappable reads in the teratocyte libraries also mapped to miRNAs/pre-miRNAs in miRBase (Supplementary Table 1). Prediction software identified a total of 35 pre-miRNAs specifically expressed in larvae, 70 specifically expressed in teratocytes, and 71 expressed in both (Fig. 1a, Supplementary Data 1). Copy number distributions indicated the most abundant predicted miRNAs in larvae and teratocytes were 21–23 nt in length (Supplementary Tables 2 and 3). More than 80% of the annotated pre-miRNAs were predicted to produce both 5p and 3p miRNAs, while the remainder were predicted to produce mature miRNAs from just one arm (Fig. 1b). More than 90% of the mature miRNAs identified in *C. vestalis* were classified as conserved or highly conserved homologs of miRNAs in miRBase, while 7% were classified as novel (Fig. 1b). Read counts normalized to the

numbers of reads in each library and expressed as reads per million (RPM) revealed highly variable abundance of miRNAs (Supplementary Data 1). In larvae, the six most abundant miRNAs were miR-1-3p, miR-184-3p, miR-275-3p-1, miR-2944, let-7, and miR-8, while in teratocytes they were miR-375-3p, miR-8, miR-184-3p, miR-275, miR-7929-3p, and miR-3405-3p (Supplementary Data 1). Read mapping to the *C. vestalis* genome showed that 71% of miRNAs were intergenic while 55% were clustered (Fig. 1b).

**CvBV-derived miRNAs are expressed in the parasitized hosts.** Like all polydnaviruses (PDVs), CvBV persists as a provirus in the genome of all *C. vestalis* cells, while replication only occurs in ovary calyx cells of females[3]. Replication results in amplification and packaging of 35 proviral DNA segments (C1-C35) into virions that infect host tissues within 1 h of egg deposition by *C.*

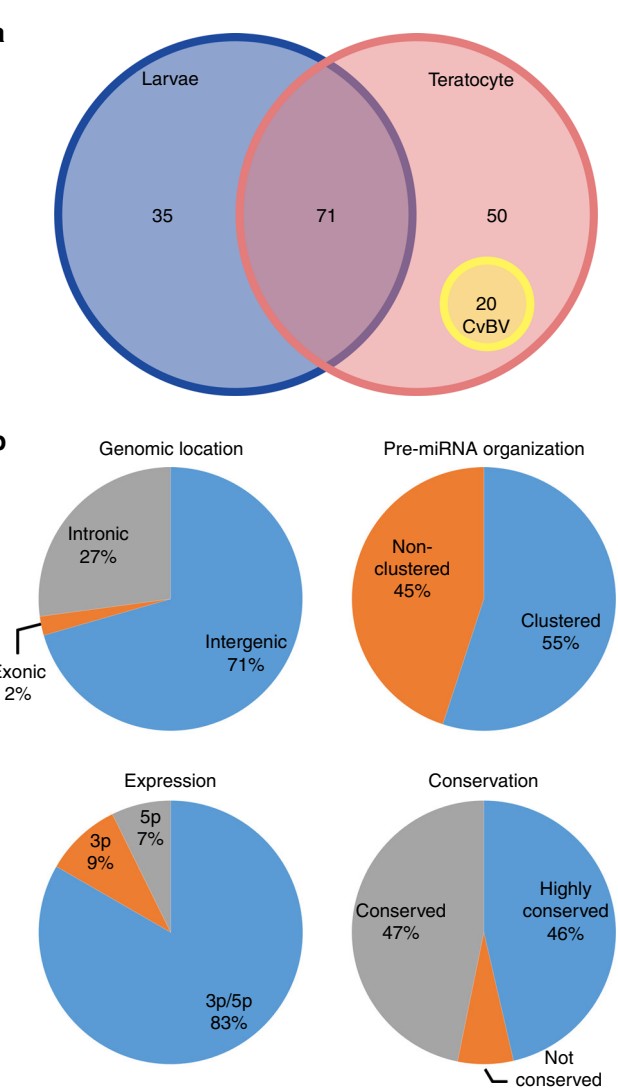

**Fig. 1** Characteristic of miRNAs in *C. vestalis*. **a** miRNAs in *C. vestalis* larvae and teratocytes. The yellow circle indicates 20 miRNA precursors expressed in teratocytes and encoded within the CvBV proviral genome. **b** Characteristics of pre-miRNAs in the genome of *C. vestalis*, mature miRNAs, and conservation with miRNAs identified in other organisms as identified in miRBase. miRNAs in which >18 nucleotides matched miRNAs in miRBase were classified as highly conserved, while miRNAs with 10–18 matching nucleotides were classified as conserved

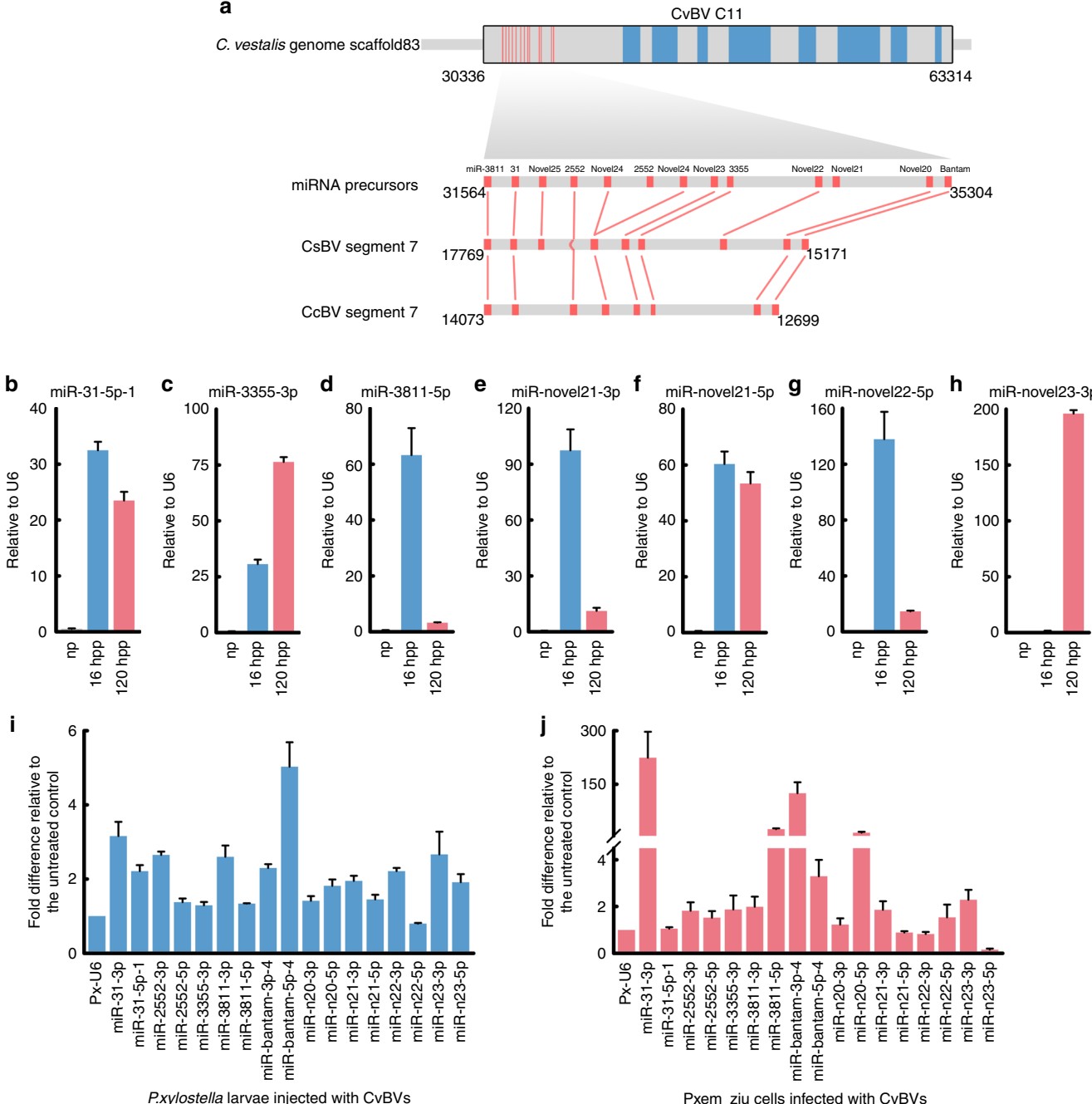

**Fig. 2** CvBV-derived miRNAs and their expression in *P. xylostella*. **a** miRNAs precursors located in CvBV segment C11. The upper bar schematically shows segment C11 in *C. vestalis* genome scaffold 83. Red vertical lines represent miRNA precursors while blue vertical bars identify predicted protein coding sequences. Below the upper bar is shown the domain of segment C11 that encodes miRNA precursors together with schematics showing miRNA precursors identified in CsBV segment 7 and CcBV segment 7 (Results). **b–h** RT-qPCR analysis of eight CvBV-derived miRNAs in *P. xylostella* normalized to *P. xylostella* U6 snRNA. np: non-parasitized *P. xylostella*, hpp: hours post parasitized. Ct values for undetected samples are considered as 40. **i** RT-qPCR analysis of 17 CvBV-encoded miRNAs in *P. xylostella* larvae injected 24 h earlier with 0.1 FEs of CvBV particles. *P. xylostella* were injected during third instars. Relative expression of CvBV-encoded miRNAs was normalized to the *P. xylostella* U6 snRNA. **j** RT-qPCR analysis of 17 CvBV-encoded miRNAs in Pxem-ZJU cells (5 × 10⁶ cells) 24 h post infection with 20 FEs of CvBV particles. The expression of CvBV-encoded miRNAs was normalized to *P. xylostella* U6 snRNA. Results shown are mean relative abundance ± s.e.m. of each miRNA from three independent biological replicates

*vestalis* adult females into *P. xylostella* larvae[13]. In contrast, *C. vestalis* eggs hatch, releasing a first instar wasp larva and teratocytes, at 28–30 h after oviposition. Mapping to the *C. vestalis* genome indicated that 20 of the miRNAs detected in teratocyte libraries are derived from 13 precursor miRNAs that are located in a 3.3 kb domain in CvBV proviral segment C11 (HQ009534.1) (Fig. 2a; Supplementary Data 1)). Each precursor miRNA formed

predicted hairpin structures (Supplementary Data 1). We also identified orthologous miRNA precursors (Supplementary Data 1) on published proviral segments of two PDVs, *Cotesia congregata* bracovirus (CcBV) and *C. sesamiae* bracovirus (CsBV)[14,15], that are carried by wasps (*C. congregata* and *C. sesamiae*) closely related to *C. vestalis*. We selected 7 of the 20 CvBV-encoded miRNAs for further study by assessing their presence in

*P. xylostella* larvae from three treatments: (1) 16 h post oviposition by *C. vestalis*, a time point when host tissues are infected by CvBV but parasitoid eggs have not hatched, (2) 120 h post oviposition when host tissues are infected by CvBV and teratocytes are also present, and (3) non-parasitized larvae (negative control (NC)). Results showed that some miRNAs were more abundant at 16 h while others were more abundant at 120 h (Fig. 2b–h). In the non-parasitized host larvae, none of these miRNAs were detected (Fig. 2b–h). These results indicated that several CvBV-encoded miRNAs emerge in parasitized host larvae through viral infection of host tissues, expression in teratocytes, or both. To assess whether CvBV-encoded miRNAs are expressed in CvBV-infected host tissues, CvBV virions were collected from female wasps as previously described[16]. The CvBV collected from the reproductive tract of a single adult female is defined as one female equivalent (FE). We then either injected 0.1 FE of CvBV particles into individual third instar host larvae or added 20 FE of CvBV particles to cultures of Pxem_ZJU cells, which is an established *P. xylostella* cell line derived from embryos. Abundance of the 20 CvBV-encoded miRNAs was then assessed in each sample by real-time quantitative PCR (RT-qPCR) (Fig. 2i and j). Primers designed to amplify miR-novel24-3p, miR-novel24-5p, and miR-novel25-5p yielded inconsistent results and were excluded from further analysis. For the remaining 17 miRNAs, all but miR-novel22-5p were detected at 24 h post oviposition at greater relative abundance than the *P. xylostella* U6 snRNA, which served as the internal normalizing control (Fig. 2i). Similar results were observed in CvBV-infected Pxem_ZJU cells, albeit relative abundance of several miRNAs was substantially higher in comparison to the U6 snRNA normalizer. Altogether, these data support the expression of CvBV-encoded miRNAs in CvBV-infected hosts.

**Teratocytes deliver miRNAs into the host.** CvBV-encoded miRNAs expressed in infected host tissues have the potential to interact directly with host gene products, whereas miRNAs expressed in *C. vestalis* teratocytes would have to enter host cells after secretion. To investigate whether miRNAs expressed in teratocytes are secreted, we collected teratocytes from parasitized *P. xylostella* larvae and placed them into primary culture followed by RT-qPCR analysis of the 30 most abundant miRNAs detected in snRNA teratocyte libraries. Primers specific to the *C. vestalis* β-tubulin were used as a control for detection of teratocyte contamination in sample media (Supplementary Fig. 2a). Ct values indicated that each miRNA was detected at greater abundance in teratocytes than in the medium used to culture teratocytes (Fig. 3a). However, the ratio of abundance for selected miRNAs including miR-34-5p and miR-275-5p-2 were less dissimilar between teratocytes and conditioned medium, suggesting that these factors are potentially secreted into the medium at higher abundance than the other most abundant miRNAs we detected in teratocytes (Fig. 3a). Since extracellular miRNAs are often associated with exosomes, we examined uptake of fluorescently labeled exosomes by the Pxem_ZJU cell line, using methods described by Bruns et al.[17] Results indicated uptake of teratocyte-produced exosomes by Pxem_ZJU cells (Fig. 3b). We then assessed whether any of the 30 most highly expressed miRNAs in teratocytes were present in Pxem_ZJU cells after co-culture with teratocytes for 6 h followed by removal of teratocytes and processing of Pxem_ZJU cells for RT-qPCR analysis. PCR analysis of *C. vestalis* 18S RNA was first used to confirm that RNA isolated from Pxem_ZJU cells after co-culture and separation from teratocytes contained no *C. vestalis* RNA (Supplementary Fig. 2b). RT-qPCR analysis then showed that 20 teratocyte-miRNAs, including miR-375-3p, miR-281-3p, and

miR-9-5p-1, were more abundant in Pxem_ZJU cells that were co-cultured with teratocytes than in control Pxem_ZJU cells that were cultured without teratocytes (Fig. 3c). We further noted that miR-375-3p, miR-281-3p, and miR-9-5p-1 are *C. vestalis* miRNAs that are not encoded in the CvBV genome.

Since many miRNAs are conserved between species, we next compared miRNAs identified from *C. vestalis* to the inventory of miRNAs previously identified from *P. xylostella* (Fig. 3d, Supplementary Data 1)[12,18]. The comparison showed that the sequence of miR-9-5p-1 in *C. vestalis* and that in *P. xylostella* are the same. In contrast, miR-281-3p and miR-375-3p from *C. vestalis* and *P. xylostella* exhibited four and six mismatches, respectively (Fig. 3d). These mismatches allowed us to design primers for distinguishing between the *C. vestalis* and *P. xylostella* variants for miR-281-3p and miR-375-3p by cloning and sequencing amplified products. In Pxem_ZJU cells co-cultured with teratocytes, sequencing of a 100 randomly selected miR-281-3p and miR-375-3p clones indicated that 77% and 87%, respectively, were the *C. vestalis* isoform (Supplementary Fig. 2c and Supplementary Data 1). The same assay using hemocytes collected from *P. xylostella* at 5 days post oviposition indicated 91% of miR-281-3p and 82% of miR-375-3p were the *C. vestalis* variants (Fig. 3d). Altogether, these results strongly suggested that miRNAs released from teratocytes were taken up by Pxem-ZJU cells and hemocytes from parasitized host larvae.

**Identification of wasp miRNA targets in the host.** We predicted candidate target genes for the 30 most abundant miRNAs expressed in teratocytes and/or encoded by CvBV using the 3′-UTRs for all *P. xylostella* messenger RNA (mRNA) genes with complete coding sequences and three prediction programs: RNA-hybrid, Targetscan, and miRanda. We classified *P. xylostella* genes as high-confidence candidate targets for a given miRNA if predicted by at least two of the prediction programs (Supplementary Data 2). A total of 91 candidate *P. xylostella* target genes were identified with six teratocytes abundant miRNAs and six CvBV-encoded miRNAs having no predicted targets. Among these host genes, two candidates were predicted to be targeted by two or more *C. vestalis* miRNAs. The targets were the *P. xylostella* ecdysone receptor (*EcR*, JN796249.1), which was the predicted target for Cve-miR-281-3p, Cve-miR-31-5p-1, and Cve-miR-novel22-5p, and prophenoloxidase 1 (*proPO 1*, GU199189.1), which was the predicted target for Cve-miR-750-3p and Cve-miR-31-5p-1. These observations suggested that teratocytes and CvBV may cooperate with each other in producing miRNAs that target the host *EcR* and *proPO 1* genes. The overall list of predicted host gene targets also suggested that *C. vestalis* miRNAs expressed in teratocytes or encoded by CvBV preferentially interact with host factors associated with immunity (prophenoloxidase and prophenoloxidase-activating proteinase 3, IMD-like protein), the nervous system (neuropeptide-like peptide, short neuropeptide F, nicotinic acetylcholine receptor alpha 6, glutamate-gated chloride channel), protein phosphorylation (protein phosphatase 2 and 4), apoptosis (caspase-1, -5, -9, apoptosis regulator buffy), and cell adhesion (cadherin-like protein and stromal cell-derived factor 2-like).

**Wasp-derived miRNAs arrest host development.** The *EcR* gene is essential for the function of the molting hormone ecdysone in *P. xylostella* larvae. Host larvae parasitized by *C. vestalis* exhibit altered growth including pronounced reductions in size and inhibition of metamorphosis[19]. We therefore focused on Cve-miR-281-3p, for which homologous miRNAs are encoded by *P. xylostella*, and Cve-miR-novel22-5p and Cve-miR-31-5p-1, for which no homologous miRNA is encoded by *P. xylostella* (Supplementary Data 1), as candidate virulence factors that target the

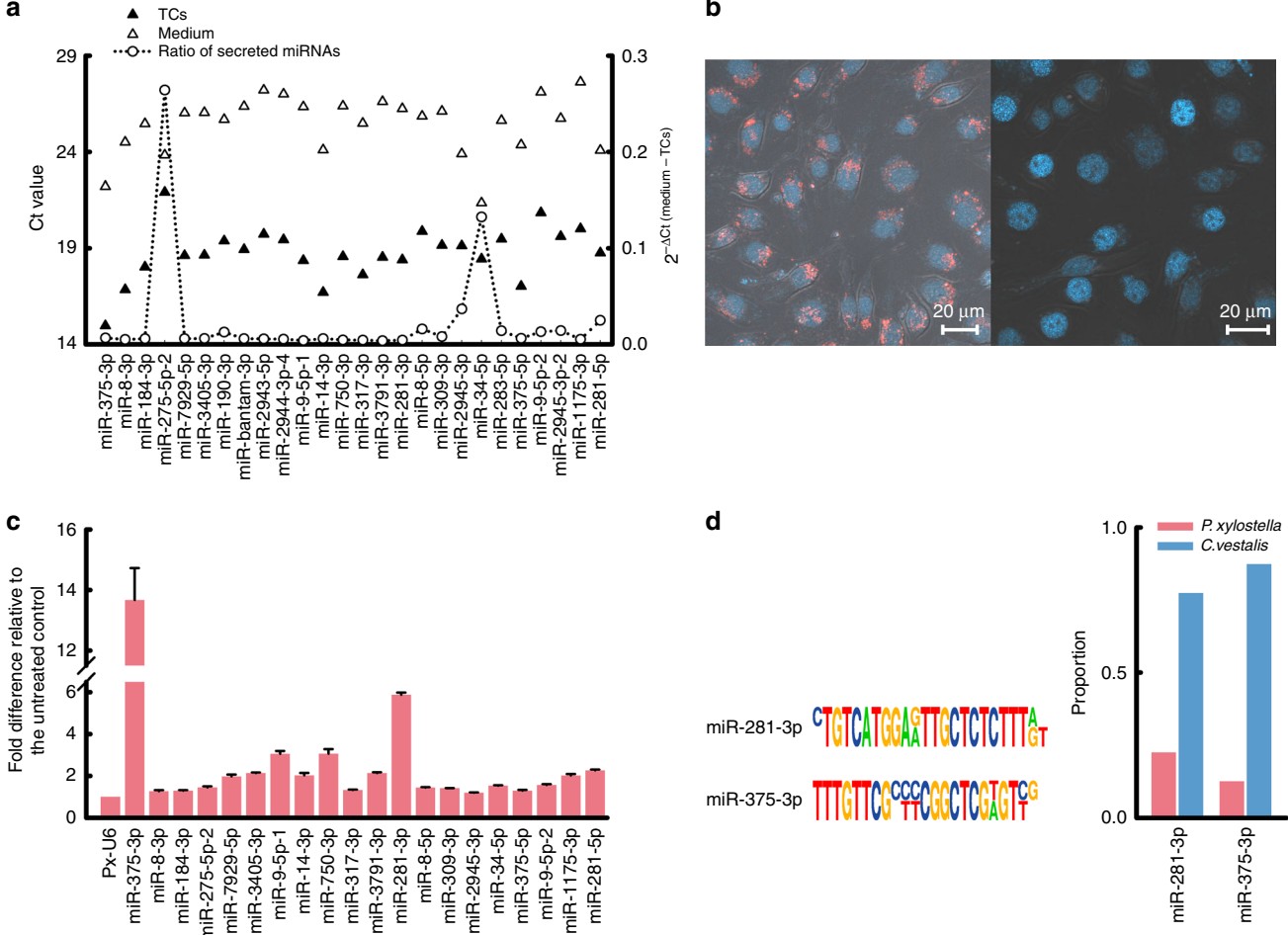

**Fig. 3** Teratocytes release miRNAs into the host. **a** Ct values (left axis) for 30 miRNAs detected in primary cultures of teratocytes and medium conditioned by teratocytes. Each primary culture contained $1 \times 10^5$ teratocytes in 1 ml of medium, which were each extracted for isolation of miRNAs after 24 h in primary culture. The abundance ratio (right axis) for each miRNA detected in the medium and in teratocytes is indicated by open circles connected with a dotted line. **b** Uptake by Pxem_cells of exosomes labeled with Alexa flour 568 (left). Pxem_cells incubation with cultured medium without fluorescent stain was used as control. Red: Alexa flour 568 NHS; Blue: DAPI. **c** Detection of the 30 most highly expressed miRNAs in teratocytes in Pxem_ZJU cells after co-culturing for 6 h. The y axis shows fold differences for each miRNA in Pxem_ZJU cells co-cultured with teratocytes versus Pxem_ZJU cells cultured without teratocytes using the *P. xylostella* U6 snRNA as the endogenous control. **d** Sequence comparison of two miRNAs between *C. vestalis* and *P. xylostella*. Sequence differences of two miRNAs between *C. vestalis* (lower line) and *P. xylostella* (upper line) are shown on the left. Proportion of two different isoforms of miR-281-3p and miR-375-3p in parasitized *P. xylostella* larval hemocytes ($n = 100$) (right). The pink columns represent the proportion of miRNAs from *P. xylostella* and the blue ones represent that of *C. vestalis* miRNAs. Primers were designed based on the miR-281-3p and miR-375-3p from *C. vestalis* and *P. xylostella*. PCR-amplified fragments were cloned and sequenced, and the difference sequence numbers were calculated. Results shown are mean relative abundance ± s.e.m. of each miRNA from three independent biological replicates

*EcR* gene and potentially alter host development (Fig. 4a). In vitro dual luciferase reporter assays using a *P. xylostella* EcR reporter construct expressed in HEK293 cells indicated that Cve-miR-281-3p or Cve-miR-novel22-5p reduced *Pxy-EcR* expression while Cve-miR-31-5p-1 did not (Fig. 4b and Supplementary Fig. 3a). To determine the effects of Cve-miR-281-3p and Cve-miR-novel22-5p on the target genes in vivo, we injected agomirs of these two miRNAs into third instar *P. xylostella* larvae. Untreated (blank control) and control miRNA mimic sequences (NC, EXIQON 479903-001, USA) were used as negative controls. We first assessed miRNA changes in host hemocytes by injecting different amounts of a Cve-miR-281-3p mimic or a negative control miRNA mimic into day 1 third instar *P. xylostella*. RT-qPCR assays showed that the abundance of Cve-miR-281-3p detected in host hemocytes 12 h post injection dose-dependently increased, whereas no increase in Cve-miR-281-3p was detected in untreated larvae (blank control) or larvae injected with a negative control

miRNA mimic sequence (Supplementary Fig. 3b). We next measured relative transcript abundance of *Pxy-EcR* during the third instar in untreated control larvae. The data showed that *Pxy-EcR* increased up to 24 h, which corresponds with a critical size and the initiation of apolysis, and then decreased to low levels through 48 h when larvae ecdysed to the fourth instar (Supplementary Fig. 3c). This pattern was consistent with studies in other insects, i.e., EcR protein increases preceding the onset of molting and then declines[20,21]. We then compared *Pxy-EcR* transcript abundance by RT-qPCR and protein abundance using an anti-EcR antibody and immunoblotting between untreated larvae or larvae injected with a control miRNA and larvae injected at the beginning of the third instar with miR-281-3p or miR-novel22-5p. Results showed that larvae injected with miR-281-3p or miR-novel22-5p exhibited lower abundance of *Pxy-EcR* or EcR protein at 24 h and then higher abundance at 36 h (Fig. 4c and Supplementary Fig. 4). This indicated that miR-281-3p or miR-novel22-5p treatment delayed

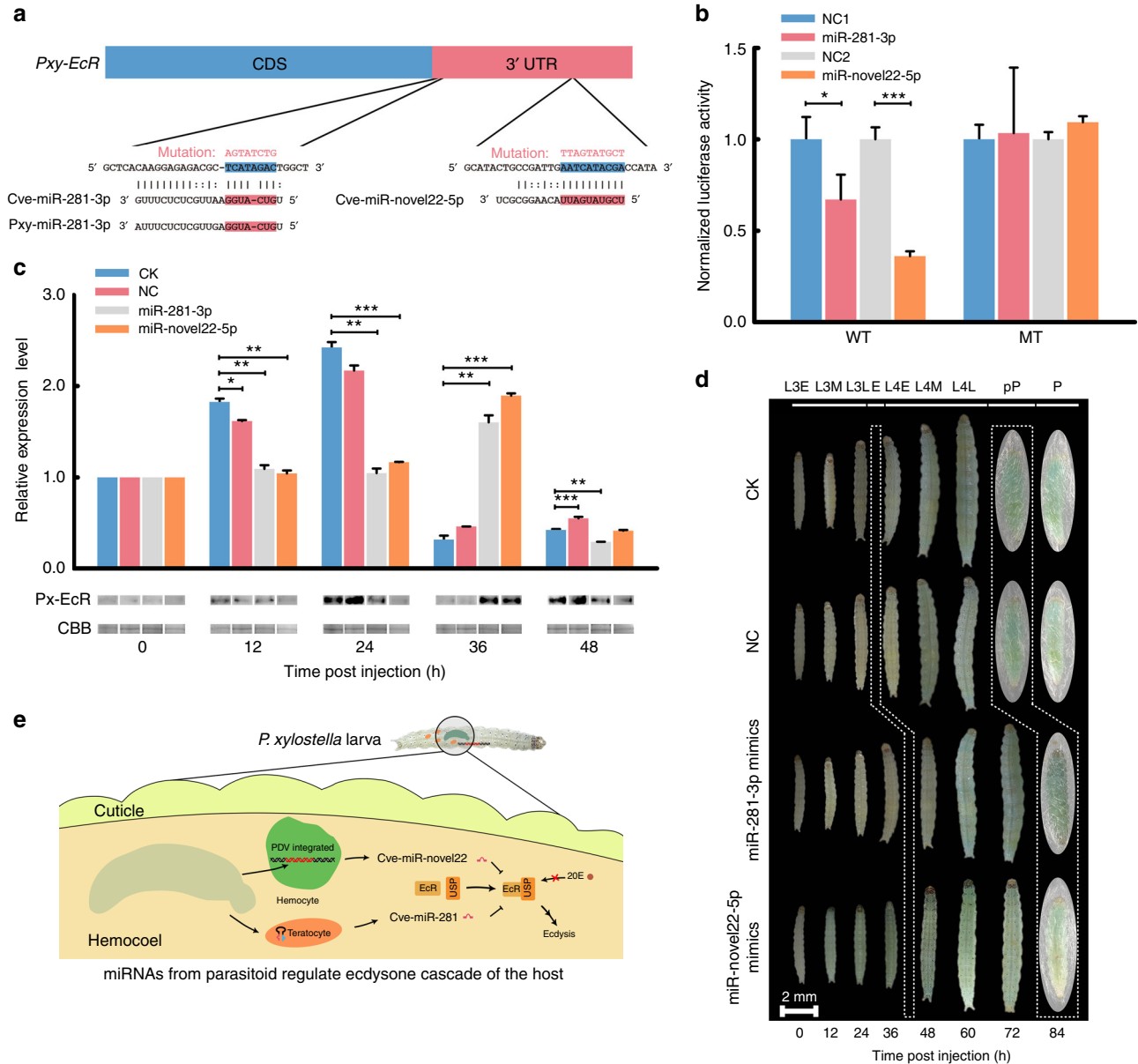

**Fig. 4** Cve-miR-281-3p and Cve-novel22-5p delay growth and pupation of *P. xylostella*. **a** Nucleotide sequences showing the complementarity of Cve-miR-281-3p/Cve-miR-novel22-5p and Pxy-miR-281-3p with the wild-type (WT) or mutant (MT) site (red letter) in *Pxy-EcR* on the Cve-miR-281-3p seed-binding region, highlighted in red. **b** *EcR* 3′-UTRs are direct targets of Cve-miR-281-3p or Cve-novel22-5p. *pMIR-REPOR*-EcR luciferase constructs, containing a WT or MT EcR 3′-UTR were transfected into HEK293 cells. Relative fluorescence ratios in HEK293 cells co-transfected with the indicated *pMIR-REPORT*-EcR luciferase vector plus either a negative control miRNA (NC1 or NC2), Cve-miR-281-3p, or Cve-novel22-5p mimic. Results shown are mean of four independent experiments ± s.e.m. Differences between groups were analyzed by Student's *t* test (*$p < 0.05$, **$p < 0.01$, ***$p < 0.001$). **c** Transcript (upper graph) and protein levels (lower immunoblots) for *Pxy-EcR* following miRNA agomir treatment. The time points of miRNA agomir injection were treated as 0 h. Transcript abundance is reported as the mean ± s.e.m. from analysis of three individuals per treatment and time point. Differences between the control and treatment groups were analyzed by Student's *t* test (*$p < 0.05$, **$p < 0.01$, ***$p < 0.001$). Coomassie brilliant blue G-250 (CBB) staining was used as a loading control. CK: untreated larvae, NC: larvae injected with a control miRNA. **d** Development of *P. xylostella* larvae following miRNA agomir treatment. Sixty larvae were tested in each treatment and their development were monitored until pupation. In the CK and NC groups, *P. xylostella* larvae initiated apolysis at 24 h post injection (the first white box), while larvae treated with Cve-miR-281-3p or Cve-miR-novel22-5p mimics initiated apolysis at 36 h. At 72 h, larvae in the CK and NC groups pupated, while Cve-miR-281-3p or Cve-miR-novel22-5p treated larvae pupated 12 h later. L3E, -3M, -3L: represent early stage, middle stage, and later stage of third larvae, respectively. E: ecdysis. L4E, -4M, -4L: represent early stage, middle stage, and later stage of 4th larvae, respectively. pP: prepupa, P: pupa. **e** Cartoon illustrates that Cve-miR-281-3p from teratocytes and Cve-miR-novel22-5p from CvBV regulate ecdysone cascade of the host

the rise of *Pxy-EcR* as observed in control larvae. Correspondingly, treated larvae exhibited a ~12 h delay in molting to the fourth instar and metamorphosis relative to controls (Fig. 4d). These results overall suggested that miRNAs secreted by teratocytes or

expressed by CvBV in infected host contribute to the developmental delays in parasitized *P. xylostella* larvae and guarantee that the host would offer sufficient resources and time for wasp offspring to complete its development (Fig. 4e).

## Discussion

Our knowledge about host control has been mostly based on functional proteins released by wasp-associated factors, such as venom, PDVs, and teratocytes. In this study, we report that *C. vestalis*-derived miRNAs are released into *P. xylostella* from teratocytes or CvBV, and that injection of two wasp-associated miRNAs delayed host molting and metamorphosis.

It has been more than a decade since the first discovery of miRNAs in Epstein-Barr virus[22]. More than 250 miRNAs have since been identified from both RNA and DNA viruses from diverse families[23]. However, no miRNAs have previously been described from PDVs. More than 70% of the encapsidated CvBV genome consists of non-coding sequence[13]. Unlike virulence genes dispersed in different PDV segments, the identified miRNAs in CvBV reside on only one segment with miRNA homologs similarly residing on a single orthologous segment of CcBV and CsBV[14,15]. Alignment with other BVs further suggests this segment is unique to BVs associated with braconid wasps in the genus *Cotesia*. The higher number of miRNAs and two duplications of miRNA precursors in CvBV further suggest that alterations have occurred in the miRNA inventory due to divergence of *C. vestalis* from *C. congregata* and *C. sesimiae*. When it comes to the function of CvBV-coding miRNAs, CvBV-derived miRNAs can be grouped into two classes: those that are analogs of host miRNAs (*P. xylostella*) and those that are bracovirus-specific. For the analogs, the seed sequence is shared with host miRNAs, whereas CvBV miRNAs have potentially evolved to target new sites in hosts or function as regulators of wasp or viral gene expression. Most known functions of virus-encoded miRNAs are involved in controlling host immune responses, prolonging longevity of infected cells and regulating the lytic cycle of infected host cells[23]. Target candidates of CvBV-derived miRNAs suggest roles in modulating host immune responses, protein neurophysiology, and development, which is consistent with the known functions of BV in parasitism of hosts by wasps[5]. In addition, it is not surprising to find that CvBV-derived genes or miRNAs expressed in teratocytes given that all wasp cells contain the CvBV proviral genome. CvBV-derived miRNA in teratocytes may function as a form of dosage compensation for CvBV or act as a signal to initiate CvBV miRNA expression.

miRNAs released by cells via extracellular vesicles (EVs, such as exosomes or microviscles) into circulation have been previously reported in studies of vertebrates in the context of serving as biomarkers for disease detection[24–28]. While the release of exocytotic vesicles by teratocytes is known[29], the production of EVs by teratocytes has not been previously reported. Previous studies have reported the secretion of several proteins by teratocytes[30,31]. Our results add to this literature by supporting that teratocytes also secrete miRNAs. Detection of several teratocyte-derived miRNAs in the medium of teratocyte primary cultures led to the hypothesis that *C. vestalis* teratocytes also likely secrete miRNAs into the hemolymph of parasitized *P. xylostella* larvae. The large number of miRNAs produced by teratocytes are also likely to regulate host in various ways due to the potential of the target candidates.

Insect growth is a complex process involving multiple signal pathway networks. Juvenile hormone and ecdysone are key regulators of insect growth, molting, and metamorphosis. In parasitoid-host interactions, larval endoparasitoids often prolong the host larval stage, reduce body size, and disrupt metamorphosis[3,6]. Recent studies also show that miRNAs are effectors of several genes with roles in hormone signaling pathways[32,33]. In *Drosophila*, bantam, miR-14, and miR-34 modulate ecdysone signaling[34–36]. In lepidopteran insects, let-7 and miR-281 also play regulatory roles in molting and metamorphosis[19,37]. miR-14 from *Drosophila* and miR-281 from *Bombyx mori* have been implicated in regulating ecdysone signaling by limiting the expression of *EcR*[19,36]. In our study, miRNAs from both teratocytes and CvBV share *EcR* as their common target, implying a role of miRNAs in host developmental regulation. We demonstrated that both teratocytes-derived Cve-miR-281-3p and PDV-derived Cve-miR-novel22-5p downregulated *EcR* gene expression and decreased the production of protein, which correlated with delayed host development. In our experiments, metamorphosis was delayed rather than arrested probably due to the injection of each host larva with only a single dose of each miRNA, whereas these factors are continuously produced in naturally parasitized hosts. Previous reports and our data jointly suggest an evolutionary conserved relationship between miR-281 family and the *EcR* in Lepidoptera. Thus, other parasitoid species are likely to take advantage of the same strategy to manipulate the development of their lepidopteran hosts. In addition, the newly functional confirmation of Cve-miR-novel22-5p from PDV indicated the convergent evolution of similar function in teratocyte and PDV, but the factors may cooperate in a successive manner to achieve the prolongation of host larval stage and in turn the completion of wasp offspring development.

To the best of our knowledge, this is the first evidence demonstrating the molecular mechanisms by which parasitoid wasps control host development via miRNA. Our findings indicate that endoparasitoids produce miRNAs with roles in altering host growth. Other miRNAs we identified may also have roles in regulating host metabolism and immune responses.

## Methods

**Insects**. *C. vestalis* and *P. xylostella* used in the study derived from cultures that had been continuously maintained for 5 years and >100 generations. *P. xylostella* was reared on cabbage grown at 25 °C, 65% relative humidity, and 14 h light: 10 h dark photoperiod. *C. vestalis* was reared by allowing adult females to parasitize *P. xylostella* third instars. Adult wasps were fed with 20% (w/v) honey solution.

**Cell lines**. The strongly adherent *P. xylostella* Pxem_ZJU cell line, which was derived from the *P. xylostella*, was maintained in 60 mm culture dish in TNM-FH culture medium plus 10% fetal bovine serum (FBS) at 27 °C under ambient atmosphere[38]. The human embryonic kidney cell 293 (HEK293) cell line was bought from Thermo Fisher and maintained at 37 °C in 5% $CO_2$ in DMEM basic medium (Gibco) medium plus 10% FBS. All cell lines were verified to be mycoplasma-free using the PCR-based mycoplasma detection kit (Kaiji, Nanjing, China). Mycoplasma testing was performed on all lines using Mycoplasma Hoechst Stain Kit (MP Biomedicals, Solon, OH, USA).

**Small RNA libraries sequencing**. Twenty second instar *C. vestalis* larvae, teratocytes from 200 parasitized *P. xylostella* larvae (see below), 20 third instar *P. xylostella* larvae, and 20 fourth instar *P. xylostella* larvae were collected separately and total RNA isolated using TRIzol reagent (Ambion, Foster City, CA, USA). Small RNAs (~18–30 nt) were isolated by the denaturing PAGE method of Lagos-Quintana et al.[39] Small RNA sequencing libraries were then constructed using the TruSeq Small RNA Library Preparation Kit (Illumina, San Diego, CA, USA). Library sequencing was performed by Illumina HiSeq 2000. miRNA genes were inferred by miRdeep2 software against the Rfam database of release 11.0[40]. Sequences that matched miRNA targets were predicted using RNAhybrid[41], TargetScan[42], and miRanda[43] with default parameters. To ensure the reliability, the genes that were predicted to be the targets by at least two algorithms were treated as the candidates for further analysis.

**CvBV particles extraction and injection**. To assess whether CvBV-encoded miRNAs are expressed in CvBV-infected host tissues, CvBV virions were collected from female wasps as previously described[16]. The ovaries of *C. vestalis* female adult wasps were dissected into prechilled PBS and the calyx was punctured individually. The calyx fluid with PBS was filtered using a 0.22 μm filter to remove cellular debris, and centrifuged at $20,000 \times g$ for 1 h, the viral particle pellet was re-suspended in PBS. The viral particles collected from one single adult female is defined as one FE. To infect host larvae, 0.1 FE CvBV particles were injected into *P. xylostella* larvae on the second day of the third instar, the expression level of CvBV-encoded miRNAs was detected 6 h post injection by RT-qPCR. To infect Pxem_ZJU cells, 20 FE CvBV particles were added to a 35 mm culture dish containing $1 \times 10^5$ *P. xylostella* Pxem_ZJU cells, the expression level of CvBV-encoded miRNAs was detected 24 h post infection by RT-qPCR.

**Teratocyte collection and culture**. Teratocytes from *C. vestalis* and *P. xylostella* hemocytes were collected using previously established methods[44]. In brief, *P. xylostella* larvae were dissected 5 days post-parasitism in culture dishes containing TNM-FH medium (HyClone, Logan, UT, USA) plus antibiotics (ampicillin and kanamycin, each at 100 µg/l). Opening the hemocoel of larvae released the *C. vestalis* teratocytes and *P. xylostella* hemocytes that were present into the medium where they settled at the bottom of the dish. After 30 min, most hemocytes had bound to the culture dish, while the much larger, nonadherent teratocytes were collected and transferred to another dish containing fresh medium. Teratocytes were further washed 5× in medium, transferred to a microfuge tube and then gently centrifuged at $500 \times g$ for 5 min. The resulting pellet consisted of only teratocytes whose abundance was estimated using a hemocytometer. To obtain the conditioned medium, $1 \times 10^5$ cells teratocytes were collected from *P. xylostella* 5 days post oviposition, and cultured in TNM-FH medium with 10% exosome-depleted FBS. After 48 h, the teratocytes were collected with the culture medium into a microcentrifuge tube and centrifuged at $500 \times g$ for 5 min, the supernatant and the teratocytes were collected separately for miRNA isolation. PCR amplification of the β-tubulin gene of *P. xylostella* was used as a marker for hemocytes while the *C. xylostella* 18S RNA gene was used as marker for contamination of hemocytes by teratocytes after purification.

**Co-culture of teratocytes with host cells**. For teratocytes co-cultured with host cells, $3 \times 10^4$ teratocytes were added to culture dishes containing $1 \times 10^5$ *P. xylostella* Pxem_ZJU cells. Two days later, teratocytes were collected and transferred to microfuge tubes as described above. Pxem_ZJU cells were collected by washing in PBS and then incubating in 0.25% trypsin (Sigma-Aldrich, St. Louis, MO, USA) for 30 min. The resulting cells were transferred to a microfuge tube and pelleted at $3000 \times g$. 18S RNA of *C. vestalis* was used to detect if there was contamination of teratocytes in *P. xylostella* Pxem_ZJU cells. Teratocytes and teratocytes cultured with *P. xylostella* Pxem_ZJU cells were used for the detection of miRNAs and functional validation, respectively.

**Immunohistochemistry**. For Alexa-NHS labeling, teratocytes were collected and cultured with water-soluble Alexa Fluor™ 568 NHS Ester (Thermo Fisher Scientific, Hudson, NH, USA) (1 µM) in TNM-FH medium with 10% FBS medium for 30 min. The medium was removed and the cells were washed with PBS three times followed by addition of fresh TNM-FH medium with exosome-depleted FBS and cultured for 6 h. The culture medium was then collected and subjected to exosome isolation using an exosome isolation reagent (Invitrogen). The Pxem_ZJU cells were adhered to an 8-chamber slide (Thermo Fisher Scientific) incubated with/ without the fluorescently labeled medium for 18 h. DAPI was added during the last 5 min. Slides were analyzed by confocal microscopy (LSM 800, Zeiss, Jena, Germany).

**Detection of miRNA sources**. To identify the sources of miRNAs, primers were designed based on the miRNAs of *C. vestalis* that did not cover the last variable nucleotides (Supplementary Table 4). Both Pxem_ZJU cells co-cultured with *C. vestalis* teratocytes and hemocytes of parasitized *P. xylostella* were collected for miRNA extraction. PCR-amplified fragments of interest were cloned into the pGEM-T Easy Vector I (Promega, Madison, WI, USA) and sequenced. For each group of miRNA homologs, 100 clones were randomly selected and sequenced. Based on sequence differences, the proportion of each miRNA originated from *P. xylostella* versus *C. vestalis* was then determined.

**Quantitative real-time PCR of miRNAs and mRNA**. miRNAs were extracted from *P. xylostella* hemocytes and Pxem_ZJU cells using the High Pure miRNA Isolation Kit (Roche, Mannheim, Germany). For RT-qPCR, total RNA was isolated using TRIzol reagent (Ambion) according to the manufacturer's instructions. The quality and concentrations of total RNAs were estimated by electrophoresis and NanoDrop 2000 spectrophotometer (Thermo Fisher Scientific). Complementary DNAs (cDNAs) were synthesized using the NCode miRNA First-Strand cDNA Synthesis Kit (Invitrogen, Carlsbad, CA, USA). The universal RT-qPCR primer was provided by this kit. The miRNA-specific forward primers were designed based on mature miRNA sequences of *C. vestalis* by changing the U in the selected miRNA sequence to T and the Tm were adjusted to range from 50 to 55 °C (Supplementary Table 4)[45]. We also confirmed that teratocyte and hemocyte templates were not contaminated by teratocytes using PCR and specific primers for the *C. vestalis* 18S RNA gene.

For mRNA detection, total RNA was isolated using TRIzol reagent (Ambion) according to the manufacturer's instructions. The quality and concentrations of total RNAs were estimated by electrophoresis and NanoDrop 2000 spectrophotometer (Thermo Fisher Scientific). First-strand cDNAs were synthesized using the ReverTra Ace qPCR RT kit (Toyobo, Osaka, Japan) according to the manufacturer's instructions. Both β-actin (GenBank Accession Number AB282645) and β-tubulin (GenBank Accession Number EU127912) genes of *P. xylostella* were used as internal controls. Primer sequences used for RT-qPCR analysis are shown in Supplementary Table 4.

The RT-qPCR analyses using the above cDNA templates and primers specific for particular miRNAs (Supplementary Table 4) were conducted using the CFX Connect real-time system (Bio-Rad, Hercules, CA, USA) and THUNDERBIRD qPCR Mix (Toyobo, Osaka, Japan). All RT-qPCR assays were performed using four internal replicates and at least three biological replicates under the following conditions: 95 °C for 60 s and 40 cycles of 95 °C for 15 s and 60 °C for 30 s. Small nuclear RNA (snRNA) U6 gene of *P. xylostella* was used as the internal control, and RT-qPCR data were analyzed using the $2^{-\Delta\Delta Ct}$ method[46] or normalized to that of U6 snRNA using the $2^{-\Delta Ct}$ method[47].

**Luciferase assays**. 3′-UTRs of target genes were obtained by 3′-RACE using specific primers (Supplementary Table 4) and the SMART RACE cDNA Amplification kit (Clontech, Mountain View, CA, USA). All PCR-amplified DNA fragments were cloned into the pGEM-T Easy Vector I (Promega) and sequenced. The 3′-UTR of *Pxy-EcR* was cloned into the pMIR-REPORT miRNA Expression Reporter Vector (Invitrogen). Mutations at the miR-281-3p seed sequence binding region were introduced at the miRNA target sites in the 3′-UTR of the *P. xylostella* ecdysone receptor gene (*Pxy-EcR*, GenBank accession number NM_001309151.1). The mutant construct was used as the negative control. Constructs were transfected into HEK293 cells using Lipofectamine 3000 (Invitrogen). Mimics of Cve-miR-281-3p, Cve-miR-31-5p-1, and Cve-novel22-5p were also transfected into the HEK293 cells. The miRCURY LNA miRNA mimic negative control (Exiqon 479903-001, Woburn, MA, USA) was used as negative control. The Dual-Luciferase Reporter Assay System (Promega) was used to measure the interaction between miRNAs and *Pxy-EcR* in each sample selected. At 48 h post transfection, the luciferase activity was measured according to the recommended protocol using a Thermo Scientific Varioskan Flash Microplate Reader (Thermo Fisher Scientific).

**miRNA agomir treatment**. Cve-miR-281-3p and Cve-novel22-5p mimics synthesized by Sangon Biotech (Shanghai, China) were injected separately into the *P. xylostella* larvae on the first day of the third instar using a glass needle and an Eppendorf Femtojet (Eppendorf, Germany) with microcontroller (Narishige, Japan). The blank control and miRCURY LNA miRNA mimic negative control (Exiqon 479903-001) were designed following a universal rule[48,49]. The expression level of miR-281-3p in *P. xylostella* haemocytes was detected every 12 h post injection by RT-qPCR, as described above. The development of *P. xylostella* in the mimic-treated group (5 pmol), negative control group (5 pmol), and non-treated group was monitored using a Keyence VHX-2000C (Keyence, Japan) microscope at 12 h intervals. Total RNA at 0, 12, 24, 36 and 48 h post injection was extracted as previously described. For immunoblotting, larvae were processed using the One Step Animal Tissue Active Protein Extraction Kit (Sangon, Shanghai, China). Samples were then run on 10% SDS polyacrylamide gel electrophoresis (PAGE) gels and transferred to Immobilon-P (Millipore) membranes. After washing, the membrane was blocked with 2% BSA followed by incubation for 2 h with *P. xylostella* EcR monoclonal antibody (diluted 1:1200). Samples were then visualized on at room temperature for 1 h. After a further washing step, immunoreactive bands were visualized on X-ray film using a horseradish peroxidase-conjugated goat anti-rabbit secondary antibody (Abcam, Eugene, OR, USA) (1:2500) and ECL detection reagents (SuperSignal West Dura Extended Duration Substrate, Thermo Fisher Scientific).

**Statistical analysis**. All statistical analyses were performed using SPSS 16.0 software. Data are presented as mean ± s.e.m. Student's *t* test was used to determine the difference between two groups. *p*-value <0.05 was considered significant.

**Data availability**. The genome assembly data have also been deposited at DDBJ/ EMBL/GenBank under accession numbers LQNH00000000. The sequencing data files for the miRNA profiling and genome information of *C. vestalis* can also accessed from the Parasitica web portal (http://www.insect-genome.com/parasitica/).

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

## Acknowledgements

We thank Drs. Francesco Pennacchio and Shu-sheng Liu for discussion and suggestions during the course of this study. This work was supported by funds from the State Key Program of National Natural Science Foundation of China (grant 31630060), the Chinese National Key Project for Basic Research (973 Project) (grant 2013CB127600), and the National Science Fund for Innovative Research Groups (grant 31321063).

## Author contributions

Conceptualization, X.x.C. and M.R.S.; methodology, X.x.C., M.S., Z.z.W., and X.q.Y.; software, F.L., Y.n.Z., D.h.G., C.l.Y., X.d.F., and B.y.Z.; validation, J.h.H., M.S., Z.z.W., X.q.Y., and Z.h.W.; investigation, Z.h.W., X.q.Y., Z.z.W., Y.n.Z., Q.j.G., X.t.W., and N.n.H.; resources, M.S., X.q.Y., S.j.W., R.m.H., L.q.Z., Q.j.G., J.n.Z., T.C., Y.p.W., and Y.n.Z.; data curation, Y.n.Z., D.h.G., and C.l.Y; writing original draft, X.x.C., Z.z.W., M.R.S., and X.q.Y.; writing review and editing, X.x.C., M.R.S., J.h.H., M.S., Z.z.W., and X.q.Y.; supervision, X.x.C.; funding acquisition, X.x.C.

## Additional information



