## [Peer Review File · Nature Communications]

Editorial Note: This manuscript has been previously reviewed at a another journal that is not operating a transparent peer review scheme. This document only contains reviewer comments and rebuttal letters for versions considered at Nature Communications.

Reviewers' Comments:

Reviewer #2:

Remarks to the Author:

Summary: This is an excellent and thorough study containing exciting results, that parasitoids deliver microRNAs (MiRs) to their hosts through their symbiotic virus and specialized cells (teratocytes), and that some of these miRs induced developmental delay in the host through modulating expression of the ecdysone receptor. I have no significant criticisms. The writing can be improved in places. My specific comments are provided below.

Abstract

L26. Define teratocytes briefly here e.g. "teratocytes (specialized wasp cells)"

L31. "The abundance of certain miRNAs produced by teratocytes in the parasitized host are mainly from teratocytes while the expression level of miRNAs encoded by CvBV can be hundredfold due to the infection of CvBV." This is a confusing sentence. Possible suggested change "Certain miRNAs in the parasitized host are mainly produced by teratocytes, while the expression level of miRNAs encoded by CvBV can be hundredfold greater than xxx."

L36. Please define EcR transcript. "reduce host" to "reduce the host"

L37. Use "delay of the host" it is not clear what "its" refers to.

Introduction

L45 It is a bit of a run-on sentence. Note comma needed at "particularly parasitic wasps,", Perhaps you want to change the wording for easier reading.

L47 change to "wasps, making them among.."

L73 "miRNAs while CvBV" to "miRNAs and CvBV"

L75. The wording here is clearer than in the abstract. You may wish to use something like this there.

L81 "larvae" to "larval"

L101. Please identify whether any of these are novel as opposed to conserved miRs.

L111. I recommend "libraries are derived"

L142. I recommend "infected hosts"

L178. "mismatches (Fig. 3d)". how many?

L213. "arrest the host development" - please remove "the"

L252. Briefly comment on why this host developmental delay could be advantageous for the developing wasps.

Reviewer #3:

Remarks to the Author:

This version of the manuscript is much more focused and interesting. I have only minor comments on the manuscript.

- In Table S2, the read counts appear very low for most of the PDV-derived miRNAs, while the RT-qPCR results in Fig. 2b-h show that they are expressed at reasonable levels relative to U6. Are the read counts shown in the table correct or the authors could explain this low read counts in the table?

Line 75: arrests should be arrest

Line 86: miRbase should be miRBase

Line 88: specifically expressed, 70 specifically expressed....

Line 180: mIR should be miR

Line 273: insert (P. xylostella) after miRNAs to clarify the host as it might be confused with C. vestalis. I presume it is P. xylostella based on identity with pxy-miRNAs shown in Table S2.

Line 274: Is it just the seed sequence? Based on Table S2, there is 100% identity of the PDV-

derived miRNAs with those of pxy-miRs.

Line 305: *B. mori* should be *Bombyx mori*.

Line 310: decrease should be decreased for consistency.

Line 342: cells lines should be cell line

Response to Reviewers

Reviewer #2 (Remarks to the Author):

Summary: This is an excellent and thorough study containing exciting results, that parasitoids deliver microRNAs (MiRs) to their hosts through their symbiotic virus and specialized cells (teratocytes), and that some of these miRs induced developmental delay in the host through modulating expression of the ecdysone receptor. I have no significant criticisms. The writing can be improved in places. My specific comments are provided below.

Response: Thank you for your comments and suggestions.

Abstract

L26. Define teratocytes briefly here e.g. "teratocytes (specialized wasp cells)"

Response: Corrected (L24).

L31. "The abundance of certain miRNAs produced by teratocytes in the parasitized host are mainly from teratocytes while the expression level of miRNAs encoded by CvBV can be hundredfold due to the infection of CvBV." This is a confusing sentence. Possible suggested change "Certain miRNAs in the parasitized host are mainly produced by teratocytes, while the expression level of miRNAs encoded by CvBV can be hundredfold greater than xxx."

Response: Changed as suggested (L29-32).

L36. Please define EcR transcript. "reduce host" to "reduce the host"

Response: We changed the wording as suggested (L35).

L37. Use "delay of the host" it is not clear what "its" refers to.

Response: We changed the wording as suggested (L37).

Introduction

L45 It is a bit of a run-on sentence. Note comma needed at "particularly parasitic wasps,", Perhaps you want to change the wording for easier reading.

Response: Sorry for the missing punctuation and the comma was added (L43).

L47 change to "wasps, making them among.."

Response: Corrected (L45).

L73 "miRNAs while CvBV" to "miRNAs and CvBV"

Response: Corrected (L71).

L75. The wording here is clearer than in the abstract. You may wish to use something like this there.

Response: Thank you and we changed the wording in the abstract.

L81 "larvae" to "larval"

Response: Corrected (L79).

L101. Please identify whether any of these are novel as opposed to conserved miRs.

Response: None of the listed miRNAs here are novel. We named these miRNAs according to their homologues in miRbase, and novel miRNAs in our study were named as miR-novelXX.

L111. I recommend "libraries are derived"

Response: Corrected (L110).

L142. I recommend "infected hosts"

Response: Corrected (L142).

L178. "mismatches (Fig. 3d)". how many?

Response: miR-281-3p and miR-375-3p from *C. vestalis* and *P. xylostella* exhibited 4 and 6 mismatches, respectively (L177-178). We added this in the text.

L213. "arrest the host development" - please remove "the"

Response: Corrected (L214).

L252. Briefly comment on why this host developmental delay could be advantageous for the developing wasps.

Response: Briefly comment was added in L252-254.

Reviewer #3 (Remarks to the Author):

This version of the manuscript is much more focused and interesting. I have only minor comments on the manuscript.

Response: Thank you for your comments and suggestion.

- In Table S2, the read counts appear very low for most of the PDV-derived miRNAs, while the RT-qPCR results in Fig. 2b-h show that they are expressed at reasonable levels relative to U6. Are the read counts shown in the table correct or the authors could explain this low read counts in the table?

Response: In Table S2 the read counts mean read counts of miRNAs in snRNA libraries of *C. vestalis* teratocytes. Most PDV-derived miRNAs show low abundant in teratocytes. Fig. 2b-h show their expression level in parasitized host. Thus, we assume that there is no conflict between these two parts.

Line 75: arrests should be arrest

Response: Corrected (L73).

Line 86: miRbase should be miRBase

Response: Corrected (L84).

Line 88: specifically expressed, 70 specifically expressed....

Response: Corrected (L86).

Line 180: mIR should be miR

Response: Corrected (L179).

Line 273: insert (*P. xylostella*) after miRNAs to clarify the host as it might be confused with *C. vestalis*. I presume it is *P. xylostella* based on identity with pxy-miRNAs shown in Table S2.

Response: Corrected (L275).

Line 274: Is it just the seed sequence? Based on Table S2, there is 100% identity of the PDV-derived miRNAs with those of pxy-miRs.

Response: Yes, it is just the seed sequence. The 100% identity mean the matched sequence (showed in the column: alignment) of PDV-derived miRNAs and those of pxy-miRs are 100% identity.

Line 305: B. mori should be Bombyx mori.

Response: Corrected (L307).

Line 310: decrease should be decreased for consistency.

Response: Corrected (L312).

Line 342: cells lines should be cell line

Response: Corrected (L344).